# Distribution of Nasal Myiases Affecting Roe Deer in Spain and Associated Risk Factors

**DOI:** 10.3390/ani15162396

**Published:** 2025-08-15

**Authors:** Néstor Martínez-Calabuig, Ceferino M. López, Ana Saldaña, José Aranha, Susana Remesar, Madalena Vieira-Pinto, David García-Dios, Pablo Díaz, Carlota Fernández-González, Pablo Díez-Baños, Patrocinio Morrondo, Rosario Panadero

**Affiliations:** 1INVESAGA Group, Department of Animal Pathology, Faculty of Veterinary, Universidade de Santiago de Compostela, Avda Carballo Calero, s/n, 27002 Lugo, Spain; nestor.martinez.calabuig@usc.es (N.M.-C.); anasaldana.ruiz@usc.es (A.S.); susana.remesar@usc.es (S.R.); d.garcia.dios@usc.es (D.G.-D.); pablo.diaz@usc.es (P.D.); carlota.fernandez.gonzalez@rai.usc.es (C.F.-G.); pablo.diez@usc.es (P.D.-B.); patrocinio.morrondo@usc.es (P.M.); rosario.panadero@usc.es (R.P.); 2Centre for the Research and Technology of Agro-Environmental and Biological Sciences (CITAB), University of Tras-os-Montes and Alto Douro (UTAD), Quinta de Prados, 5000-801 Vila Real, Portugal; j_aranha@utad.pt; 3Institute for Innovation, Capacity Building and Sustainability of Agri-Food Production (Inov4Agro), University of Tras-os-Montes and Alto Douro (UTAD), Quinta de Prados, 5000-801 Vila Real, Portugal; 4Department of Forestry Sciences and Landscape Architecture (CIFAP), University of Tras-os-Montes and Alto Douro (UTAD), Quinta de Prados, 5001-801 Vila Real, Portugal; mmvpinto@utad.pt; 5Instituto de Biodiversidade Agraria e Desenvolvemento Rural (IBADER), Campus Terra, s/n, 27002 Lugo, Spain; 6Veterinary and Animal Research Centre (CECAV), University of Tras-os-Montes and Alto Douro (UTAD), Quinta de Prados, 5001-801 Vila Real, Portugal; 7Associate Laboratory for Animal and Veterinary Sciences (AL4AnimalS), University of Tras-os-Montes and Alto Douro (UTAD), Quinta de Prados, 5001-801 Vila Real, Portugal

**Keywords:** *Cephenemyia stimulator*, *Oestrus ovis*, Oestridae, *Capreolus capreolus*, roe deer, myiasis, Spain

## Abstract

This study aims to understand the current situation of nasal myiases affecting roe deer in Spain and to analyze the factors that may influence its spread throughout the country. The results will be useful for identifying the most suitable measures for its control. A wide range of samples (*n* = 1600) were analyzed to determine the most important variables affecting the distribution of these myiases. Two nasal bot flies were identified, with *Cephenemyia stimulator* (around 40%) being much more prevalent than *Oestrus ovis* (around 2%). The high prevalence of *C. stimulator* in Spain and its notable increase in recent years demonstrate that preventive and control measures must be taken. Our results reveal that preventing roe deer overpopulation, especially in northern areas of Spain, would be very useful for decreasing the likelihood of nasal bot fly infestation.

## 1. Introduction

Roe deer (*Capreolus capreolus*) is the smallest and most abundant wild ungulate on the European continent [1]. Currently, this species is distributed throughout all autonomous communities of Peninsular Spain, occupying 59% of the geography, although it is mostly located in the central–northern half of the country [2].

*Cephenemyia* is a genus of flies in the family Oestridae found exclusively in the Holarctic region [3]. These insects parasitize cervid hosts, specifically those belonging to the subfamilies Cervinae and Capreolinae. In the Palearctic region, four species of *Cephenemyia* have been identified: *C. auribarbis* (Meigen, 1824), *C. stimulator* (Clark, 1815), *C. trompe* (Modeer, 1786), and *C. ulrichii* (Brauer, 1863). Among these species, *C. stimulator* is specific to roe deer [3,4,5]. Adults deposit first-stage larvae (L1) directly onto the nostrils of the animals that quickly migrate into the nasal cavity, where they remain in diapause during the cold winter months. When temperatures begin to rise in spring, the L1s resume their lifecycle, moving to the pharyngeal region. There, they enter the mucosal recesses where they molt into stage 2 and 3 larvae (L2 and L3). Once development is complete, L3s emerge through roe deer’s sneezes or coughs, burying themselves in the leaf litter, as they are photophobic, where they undergo pupation. After about 2–3 weeks, the imago emerges, and after copulation, the females search for new roe deer to parasitize [6].

It has been demonstrated that *C. stimulator* infestations have a noticeable negative effect on the health of roe deer. All larval stages have hooked mouthparts and a body covered in spines, which cause irritation and erosion of the nasopharyngeal mucosa of the roe deer. Likewise, the most noticeable signs in roe deer appear during spring and summer when the highest populations of L2 and L3 are found [7]. The presence of 30–80 mature larvae can cause serious problems for the health of the roe deer and can be fatal in young or immunocompromised animals [7,8]. Furthermore, some mature L3s can occasionally become trapped in the nasal cavity, where they die and decompose, causing a purulent focus around them [9].

*Oestrus* is another genus of the Oestridae family whose larvae develop in the nasal cavity and frontal sinuses of domestic and wild Bovidae. Currently, four species are recognized: *O. aureoargentatus* (Rodhain & Bequaert, 1912), *O. caucasicus* (Grunin, 1948), *O. ovis* (Linnaeus, 1758), and *O. variolosus* (Brauer, 1863) [3]. Among them, *O. ovis* stands out for being the most common in sheep and goats [10]. It is worth noting that its host specificity is lower than that of other oestrids since it has been found in other wild ruminants [4,11,12,13] as well as occasionally in humans and carnivores [14,15,16].

To our knowledge, epidemiological studies of *C. stimulator* in Europe were first performed in roe deer from Poland, Germany, and Hungary in the 1970s [17,18,19]. In Spain, Notario and Castresana [20] first reported the parasite in several roe deer imported from France in the province of Ciudad Real (Central Spain) in the spring of 1997. The first infestations in native roe deer were detected in Asturias and Galicia in the early 2000s [21]. In 2022, another nasal myiasis caused by *O. ovis* was detected for the first time in a roe deer from Guadalajara, Spain [13].

Since the first case of infestation by *C. stimulator* in roe deer in Spain, this myiasis has experienced such a rapid expansion [5,22,23,24,25] that it has sometimes been associated with a decrease in roe deer populations in some areas [26,27]. Due to this, a comprehensive epidemiological study was conducted, including a large number of roe deer from all its geographical distribution range in Spain, to assess the distribution of nasal myiases. Additionally, a risk factor analysis was performed to identify factors that significantly influence the prevalence and parasite burden of *C. stimulator*. The results obtained in this study will be essential for designing the most appropriate management and control measures against bot flies to limit their impact on roe deer populations.

## 2. Materials and Methods

### 2.1. Area of Study and Sampled Animals

Between February 2018 and May 2025, a total of 1600 roe deer heads from peninsular Spain (southwestern Europe) were examined. This area measures 493,518 km^2^ and comprises 15 autonomous regions (ARs) [28]. Figure 1 shows the geographic distribution of the collected samples that were obtained from regions with stable roe deer populations, which are primarily the northern half of the Iberian Peninsula. In the south, roe deer tend to be concentrated almost entirely in the province of Cádiz [2]. Animals from all Spanish ARs were included in the present study except those from the region of Murcia, where the roe deer population is very small [2,29]. Although some of the animals examined in this study (*n* = 488) were included in previous research on the prevalence of *C. stimulator* and *O. ovis* in roe deer from the northern (*n* = 304) and central (*n* = 184) regions of the Iberian Peninsula [24,25], they were also included in the present investigation due to their relevance for identifying the risk factors associated with the presence of *C. stimulator*.

Most of the animals included in this study (75%) were culled by hunters according to game laws in each AR; no animals were culled for the purpose of this study. Females were primarily targeted in late autumn and winter, whereas males were mainly culled in spring and summer, which explains the higher number of samples collected during the winter season. The remaining 25% of the animals were collected by personnel from Wildlife Recovery Centers in different regions after being found dead, mostly roadkill. These samples were received throughout the year.

Roe deer heads were preserved either fresh or frozen (−20 °C) until analysis.

### 2.2. Larval Collection and Identification

For larvae collection, the skin of the head was removed, the pharyngeal region was accessed, and the nasal cavity was opened and carefully examined for recovering all visible larvae. Afterwards, the pharyngeal cavity was washed with pressurized water on a mesh with a pore diameter of 150 μm. This last step is essential to ensure the collection of all larvae, especially L1, ranging from 1 to 3 mm in size. The mesh was observed under a stereomicroscope at 16×, and the larvae were classified by larval stage (L1, L2, and L3) and subsequently morphologically identified according to the key proposed by Zumpt [4]. In addition, the number of partially disintegrated larvae or those showing signs of putrefaction around them was recorded.

### 2.3. Roe Deer Data Collection

The animals were classified by sex and age. Four age groups were established using Hoye’s dental criteria for age estimation [30]: fawn (0–3 months); young (>3 months to 2 years); adult (>2 years to <6 years); and old (≥6 years). In addition, information about the month, season, year, and AR of collection was recorded (Table 1).

Since a variety of habitats and climates were described in Spain, the animals were also classified according to the bio-regions defined by the Spanish Wildlife Disease Surveillance Scheme (Internal report to the Spanish Ministry of Agriculture, MARM, and spatial aggregation of wildlife, 2008) [31]. Five bio-regions were considered as previously described Muñoz et al. [32]: Atlantic (Atlantic climate with high rainfall), Northern Plateau (continental mediterranean climate, dominated by rainfed agrosystems), South Central (continental thermo-mediterranean climate), Interior Mountains (highly continental mediterranean climate), and South and East Coast (locally arid temperate climate).

Another variable considered in the study was the land cover, based on the CORINE Land Cover (CLC) 2018 project [33]. Accordingly, two main categories were considered: agricultural areas (composed of arable land, permanent crops, pastures, and heterogeneous agricultural areas) and forests. The forests were subdivided into four categories: broadleaf, coniferous, mixed forested areas, and shrubs (Figure 2).

Finally, roe deer abundance was also considered as a variable using previously described predictive models of roe deer abundance [34], which were based on data from wildlife–vehicle collisions.

### 2.4. Statistical Analysis

*Cephenemyia stimulator* prevalence was calculated for each of the categories of the explanatory variables studied. Statistical uncertainty was assessed by estimating the 95% confidence interval (CI) for each of the proportions according to the binom.test() function. The possible influence of the different factors studied (Table 1) on the *C. stimulator* prevalence was analyzed using logistic regression. All explanatory variables introduced in the logistic regression were categorical variables (sex, age, month, season, year, AR, bio-region, land cover, and roe deer abundance). The final model was selected as the one with the lowest Akaike’s Information criterion (AIC) value from all of the models performed. This analysis was performed using the glm() function in the R statistical package [35]. Odds ratio values were computed by raising ‘e’ to the power of the logistic coefficient over the reference category. The possible influence of different variables on *C. stimulator* larval count was assessed using a multivariate ANOVA with R aov() function; only positive animals (*n* = 638) were included, and the natural logarithm of larval count plus 1 was used as the dependent variable. Factors were eliminated step by step forward and backwards based on the AIC values using the step() function until the best model was obtained. Pairwise analyses were performed using the Tukey’s adjustment method.

### 2.5. QGis Project

We utilized the open-source Geographical Information System software QGIS (version 3.28.6-Firenze, code revision 868c9fa03b) alongside ArcGIS 9.7 (commercial) to develop a GIS project aimed at mapping the geographic locations of samples and visualizing the spatial distribution of variables under study, specifically land cover. By leveraging geostatistical routines integrated within these GIS platforms and based on *C. stimulator* prevalence data, we generated a continuous prevalence map using the Inverse Distance Weighted (IDW) interpolation method.

## 3. Results

### 3.1. Overall Prevalence and Larval Mean Intensity

Of the 1600 roe deer analyzed in the study, 638 (39.9%; CI 95% 37.46–42.32) presented *C. stimulator* larvae (*n* = 28,120) (Figure 3a). This species was detected in all AR except Andalucía, Comunidad de Madrid, and Comunidad Valenciana. The mean larval intensity was 44.08 (SD 71.99). In addition, 26 *O. ovis*-positive animals were detected (1.6%; CI 95% 1.02–2.28) in four Spanish ARs (Aragón, País Vasco, Castilla y León, and Castilla-La Mancha) (Figure 3b). The mean larval intensity was 2.6 (SD 1.92). The total number of larvae found was 67.

It is worth noting that a single animal from País Vasco harbored both *C. stimulator* and *O. ovis* larvae: one dead L3 of *C. stimulator* in the maxillary sinus and one L1 of *O. ovis* in the nasal cavity.

Moreover, 13.2% (84/638) of the roe deer showed dead L3s trapped in the maxillary sinus (Figure 4). The number of dead and partially disintegrated larvae found per animal ranged from one to four. Although partially degraded, their chitinous cuticle preserved enough structure for morphological identification.

### 3.2. Cephenemyia stimulator Risk Factor Analysis

Nine different variables were considered for the study (Table 1). The prevalence of *C. stimulator* varied between 0% and 90%, while its larval intensity ranged from 0 to 82.2.

**Table 1 animals-15-02396-t001:** Variables studied to analyze their possible influence on the prevalence and intensity of larvae of *C. stimulator*.

Variable	Categories	Positive Animals/Total (%; CI 95%)	Mean Larvae Intensity (SD)
Sex	Female	374/1070 (35.0%; 32.09–37.90)	39.9 (58.31)
	Male	264/530 (49.8%; 45.47–54.15)	50.0 (87.58)
Age	Fawn (0–3 mo)	0/30 (0%; 0–11.57)	0
	Young (>3 mo–2 yr)	209/489 (42.7%; 38.31–47.26)	72.9 (108.45)
	Adult (>2–6 yr)	323/838 (38.5%; 35.23–41.93)	29.1 (36.34)
	Old (≥6 yr)	106/243 (43.6%; 37.29–50.11)	32.8 (40.23)
Month	January	69/138 (50%; 41.38–58.62)	50.1 (96.47)
	February	160/630 (25.4%; 22.04–28.99)	33.5 (40.50)
	March	46/68 (67.7%; 55.21–78.49)	54.2 (83.92)
	April	83/156 (53,2%; 45.06–61.23)	42.2 (52.00)
	May	48/116 (41.4%; 32.31–50.90)	41.2 (44.13)
	June	30/71 (42.3%; 30.61–54.56)	23.8 (26.56)
	July	37/68 (54.4%; 41.88–66.55)	37.4 (36.27)
	August	21/40 (52.5%; 36.13–68.49)	53.7 (61.60)
	September	37/61 (60.7%; 47.31–72.93)	31.4 (32.50)
	October	55/77 (71.4%; 60.00–81.15)	82.2 (148.65)
	November	27/81 (33.3%; 23.24–44.68)	54.2 (91.02)
	December	25/94 (26.6%; 18.00–36.71)	38.8 (48.20)
Season	Winter	275/838 (32.8%; 29.64–36.11)	41.6 (67.25)
	Spring	159/340 (46.8%; 41.36–52.22)	38.3 (46.25)
	Summer	82/154 (53.2%; 45.05–61.32)	40.1 (43.26)
	Autumn	122/268 (45.5%; 39.45–51.69)	60.0 (112.57)
Year	2018	7/14 (50%; 23.04–76.97)	21.4 (10.47)
	2019	14/26 (53.9%; 33.37–73.41)	19.9 (24.71)
	2020	20/38 (52.6%; 35.82–69.02)	26.4 (30.00)
	2021	61/96 (63.5%; 53.09–73.13)	54.6 (64.01)
	2022	126/262 (48.1%; 41.90–54.32)	56.2 (72.71)
	2023	142/425 (33.4%; 28.94–38.12)	46.3 (82.29)
	2024	215/580 (37.1%; 33.13–41.14)	38.4 (77.71)
	2025	53/159 (33.3%; 26.07–41.23)	36.4 (36.16)
Autonomous region	Andalucía	0/10 (0%; 0–30.85)	0
	Aragón	45/187 (24.1%; 18.13–30.84)	17.2 (24.07)
	Cantabria	71/125 (56.8%; 47.64–65.63)	29.8 (34.07)
	Castilla y León	220/520 (42.31%; 38.02–46.68)	39.5 (43.59)
	Castilla-La Mancha	4/314 (1.27%; 0.35–3.23)	11.5 (18.43)
	Cataluña	37/93 (39.8%; 29.78–50.46)	16.7 (17.07)
	Comunidad de Madrid	0/1 (0%; 0–97.50)	0
	Comunidad Foral Navarra	9/14 (64.3%; 35.14–87.24)	20.1 (24.76)
	Comunidad Valenciana	0/15 (0%; 0–21.80)	0
	Extremadura	1/2 (50%; 1.26–98.74)	16.0 (0)
	Galicia	148/195 (75.9%; 69.27–81.72)	79.1 (122.72)
	La Rioja	42/51 (82.4%; 69.13–91.60)	32.2 (28.90)
	País Vasco	52/63 (82.5%; 70.90–90.95)	44.2 (66.74)
	Principado de Asturias	9/10 (90%; 55.50–99.75)	36.0 (33.70)
Bio-region	Atlantic	260/368 (70.7%; 65.71–75.26)	59.2 (99.49)
	Northern Plateau	331/734 (45.1%; 41.45–48.78)	33.4 (41.18)
	South Central	1/31 (3.2%; 0.08–16.70)	16.0 (0)
	Interior Mountains	29/383 (7.6%; 5.13–10.70)	46.9 (46.56)
	South and East coast	17/84 (20.2%; 12.25–30.41)	17.3 (20.73)
Land cover	Agriculture	158/538 (29.4%; 25.55–33.42)	33.3 (41.10)
	Broadleaf	128/355 (36.1%; 31.06–41.29)	44.3 (55.28)
	Coniferous	68/153 (44.4%; 36.42–52.69)	42.7 (47.82)
	Mix forested areas	147/214 (68.7%; 62.02–74.84)	53.4 (92.70)
	Shrubs	137/340 (40.3%; 35.04–45.72)	47.0 (94.24)
Roe deer abundance	0–0.3	1/20 (5%; 0.13–24.87)	5.0 (0)
	0.3–0.7	77/235 (32.8%; 26.80–39.17)	43.7 (111.10)
	0.7–1.4	213/684 (31.1%; 27.68–34.76)	33.9 (40.82)
	1.4–2.6	186/407 (45.7%; 40.78–50.68)	55.7 (90.18)
	2.6–9.8	161/254 (63.4%; 57.14–69.32)	44.5 (53.51)

Logistic regression identified six significant predictors of *C. stimulator* prevalence: month, year, AR, combination of bio-regions with age, land cover, and roe deer abundance (Table 2).

Regarding the sampling month, the prevalence was significantly higher in January, April, June, July, September, and October compared to February, when the lowest prevalence was recorded. In addition, animals sampled during 2018, 2019, 2022, 2024, and 2025 showed higher prevalence compared to 2023.

Using the AR of Castilla-La Mancha as a reference, the prevalence was statistically higher in the communities of the Principado de Asturias, País Vasco, La Rioja, Galicia, Comunidad Foral de Navarra, Cantabria, Castilla y León, and Cataluña, ordered from highest to lowest prevalence. Figure 5 presents the spatial distribution of prevalence throughout peninsular Spain. This map achieved an accuracy of approximately 81.5%, thereby supporting the robustness of our research findings.

Regarding the combination of the factors of bio-regions and age, the prevalence was significantly higher in young, adult, and old animals from the Northern Plateau climate region and in old animals from the Atlantic region, using the Interior Mountain area + adult age as a reference. We also observed a significantly higher prevalence in roe deer inhabiting forested areas (broadleaf, coniferous, mixed forest, and shrubs) compared to those living in agricultural areas. Finally, the prevalence increased progressively from areas of low roe deer density (0–1.4) to those showing a greater abundance (1.4–9.8).

Finally, four variables were identified as significant risk factors for individual larval burden (Table 3). In this sense, a significantly higher larval load was observed in males compared to females. According to the Tukey HSD pairwise test, significantly higher larval intensities were found in young animals compared to adult (*p* < 0.001) and old animals (*p* < 0.001). In addition, significantly higher larval intensities were found in 2021 compared with 2019 (*p* = 0.040), 2020 (*p* = 0.037), and 2024 (*p* = 0.004), and in 2022 compared with 2024 (*p* = 0.002). Finally, higher larval burdens were found in Galicia than in Aragón (*p* < 0.001) and Cataluña (*p* = 0.012) and in Castilla-Leon than in Aragon (*p* < 0.001) and Cataluña (*p* = 0.015).

## 4. Discussion

To the best of our knowledge, this study represents the most extensive investigation of nasal myiases in roe deer in Europe, covering a broad area. Our results show a total prevalence of 39.9%, higher than that obtained by other authors during this century in other European countries such as Hungary, Poland, Czech Republic, Croatia, and the Netherlands, with prevalences ranging from 9.2 to 34.6% [36,37,38,39,40,41,42]. Only two studies carried out in limited areas of Slovakia (*n* = 202) [43] and Estonia (*n* = 38) [44] showed higher prevalences (40.1 and 74%, respectively). Focusing on Spain, the prevalence was higher than that obtained by Fidalgo et al. [22] in different ARs (16.5%) and Pajares et al. [45] in the northwest of the country (31.6%).

Risk analysis revealed that the prevalence obtained in the month of February was significantly lower than that registered in other months (Jan, Apr, Jun, Jul, Sep, Oct). However, these results could have been largely influenced by the geographical origin of the samples, since most animals sampled during that month came from a hunting activity organized by the Spanish Roe Deer Association in Castilla-La Mancha, where the prevalence of *C. stimulator* was less than 2%, for four consecutive years (2021–2025). Likewise, 2023 was the year with the lowest prevalence, largely influenced by a greater number of samples received from the central-southern region of the Peninsula where the observed prevalence was low.

Concerning the ARs, those located in northern Spain (Principado de Asturias, País Vasco, La Rioja, Galicia, Comunidad Foral de Navarra, Cantabria, Castilla y León, and Cataluña) presented higher prevalences (39.8–90%) compared to those from central-southern Spain (1.3%), which coincides with a previous study that found higher prevalences in northern regions than in the central peninsula [22]. However, it is important to point out that Spain shows significant climatic, topography, and vegetation cover differences among regions, which may be the underlying factors driving these patterns. Some of these variables will be discussed below.

With respect to the bio-regions, higher prevalence was observed in the Atlantic region (70.7%) and the Northern Plateau (45.1%). These two areas are mainly located in the northern part of the Iberian Peninsula, largely coinciding with the ARs showing the highest prevalences. In line with our results, Morrondo et al. [46] and Arias et al. [47] also observed higher prevalences of *C. stimulator* in the Oceanic climate (northern zone) with respect to the Mediterranean areas. However, in our study, the statistical model only considered this variable when combined with age. The model revealed the existence of significant differences in *Cephenemyia* spp. prevalence between those bio-regions where a high number of animals was sampled but, in some cases, only for certain age groups. In this sense, significantly higher prevalences were observed between the old animals sampled in the Atlantic bio-region and young, adult, and old animals from the Northern Plateau when compared to the adult animals from the Interior Mountain bio-region. This reinforces the hypothesis that certain areas of Spain could be more appropriate for the maintenance of the parasite while also suggesting that there might be an age-related influence in these areas. These results are consistent with those of Fidalgo et al. [22], who detected higher prevalences in communities in the Atlantic region compared to other regions of Spain.

Regarding land cover, significantly higher prevalences were observed in forest compared to agricultural zones, which could be explained by a combination of factors that make these areas favorable for the parasite’s development. Morellet et al. [48] observed that forest zones with more stable temperature and humidity conditions were the most frequently selected by roe deer, as the species tends to prefer habitats with vegetation cover where it can find food and better shelter, along with less human activity. In open agricultural areas, the movement of roe deer may be more dispersed, and the survival of the parasite may decrease due to greater human activity and the use of machinery and chemicals that may alter the ecosystem. Furthermore, forest areas provide refuge from predators such as the wolf, which is widely present across the northern Iberian Peninsula, where roe deer represent a significant component of its diet [49]. However, Kiraly and Egri [36] found no significant differences in *C. stimulator* prevalence between open and forested areas in Hungary, which could indicate that the differences observed in the present investigation are mainly associated with the distribution of these kinds of ecosystems throughout Spain.

Among the factors that significantly influence the prevalence of *C. stimulator*, the abundance of roe deer stands out. Increased prevalence was observed in areas with higher host densities, which coincides with Alcaide et al., who stated that *O. ovis* seroprevalence in sheep was higher in areas with greater ovine density [50]. These findings reinforce the idea that proper population management must be implemented to prevent certain density-dependent diseases from spreading through wildlife, as these could lead to a deterioration in animal welfare and population declines [51].

Regarding larval intensity, this study shows mean intensity levels (44.1) higher than those reported from other European studies carried out during this century in Hungary, Poland, the Czech Republic, Slovakia, and Estonia, with averages between 1 and 25.4 larvae/animal [36,37,38,39,40,43,44]. Previous studies carried out in northern Spain also obtained lower intensities, with larval averages of between 16.7 and 24.3 [23,52].

Considering the statistical results, significantly more larvae were detected in males (50.0%) compared to females (39.9%). These observations are in line with those previously reported by Király and Egri [36] and Pajares [5]. This could be because males expend considerable energy during the mating season, which leads to a relapse of their immune system, favoring the proliferation of parasites. Likewise, during the period of antler formation, they have high energy requirements, decreasing the animal’s resistance [36].

Furthermore, we observed a significantly higher intensity in young animals (72.9) compared to adult (29.1) and old animals (32.81). A similar pattern has been observed by other authors who described the existence of a partially protective response that caused the number of larvae to be lower in reinfestations, whereas primoinfested animals are the most parasitized. Likewise, younger animals could have a lower defense reaction against flies, allowing larviposition [36,53,54]. However, fawns that spend extended periods hidden and camouflaged within the vegetation are less exposed to flies [52], which may explain why all of them were negative for *C. stimulator*.

On the other hand, the higher larval intensities obtained in 2021 compared to 2019, 2020, and 2024 may be because during that year, the number of samples analyzed came almost entirely from the northern peninsula. The same is true between 2022 and 2024, since during the latter year, a greater number of samples were analyzed from the central and eastern peninsula, where parasite loads are not as high.

Considering the statistical results, the larval intensity was higher in the northern communities compared to southern and eastern communities, where *C. stimulator* prevalence is also low. This could be related to the infestation pressure in these areas, as the higher the number of flies, the greater the possibility of infestation and reinfestation [5,55].

A considerable number of roe deer harbored decomposing *C. stimulator* L3s in their maxillary sinuses. This finding supports the hypothesis of Fidalgo et al. (2023) [9], who suggested that L1 larvae could reach the maxillary sinus through the nasomaxillary opening and, once developed into L3s, become trapped due to their inability to leave the sinus. The presence of these larvae could cause sinusitis with secondary fungal or bacterial complications [9].

Regarding *O. ovis*, there is another report of this species in roe deer, also from Spain [56]. The prevalence obtained in the referred study (5.3%) was higher than the obtained in the present study (1.6%), which could be due to the larger sheep livestock population in that area, increasing the risk of cross-infections. The low *O. ovis* prevalence and larval intensity found suggest that the infection could be a sporadic event.

This is the first reported case of both *O. ovis* and *C. stimulator* larvae being found in a single animal. However, a simultaneous infestation cannot be confirmed since the only *C. stimulator* specimen detected was an L3 trapped in the maxillary sinus, possibly resulting from a previous infestation. However, it is highly likely that mixed myiasis may occur in areas with a high density of roe deer and sheep.

Given the impracticality of pharmacological treatment for free-ranging animals, effective population management, like regulated selective hunting, is crucial to control the spread of oestrid infestations.

## 5. Conclusions

*Cephenemyia stimulator* infestation in roe deer in the Iberian Peninsula is clearly linked to geographical and demographic factors. Northern regions of the peninsula show higher prevalence due to high roe deer density and abundant forest and shrubland habitats. Additionally, increased prevalence was observed in areas with higher host densities, reinforcing the idea that proper population management is necessary to prevent the spread of oestrid flies.

Larval intensity appears to be influenced by several variables, including sex, age, year of sampling, and autonomous region, which could be explained by hormonal, immunological, and behavioral factors and infection pressure.

This is the first reported case of both *O. ovis* and *C. stimulator* larvae being found in a single animal. The detection of *O. ovis* supports the hypothesis that the risk of cross-infection is higher in areas with dense sheep farming.

The rising incidence of nasal myiases and the impracticality of pharmacological treatment underscore the urgent need for roe deer population control to mitigate further parasitic proliferation.

## Figures and Tables

**Figure 1 animals-15-02396-f001:**
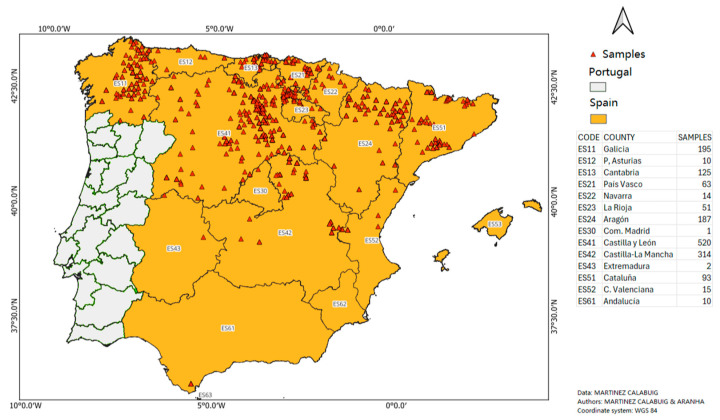
Location of the roe deer (*Capreolus capreolus*) examined.

**Figure 2 animals-15-02396-f002:**
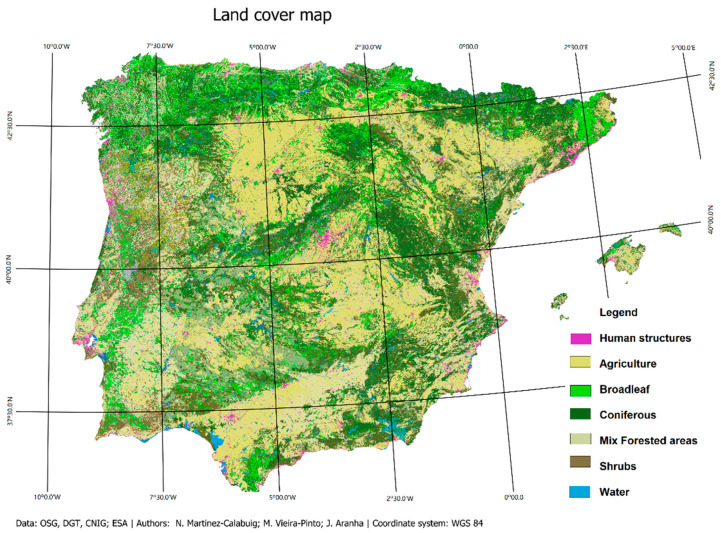
Land cover map of the Iberian Peninsula (based on the CORINE Land Cover 2018 project) [33].

**Figure 3 animals-15-02396-f003:**
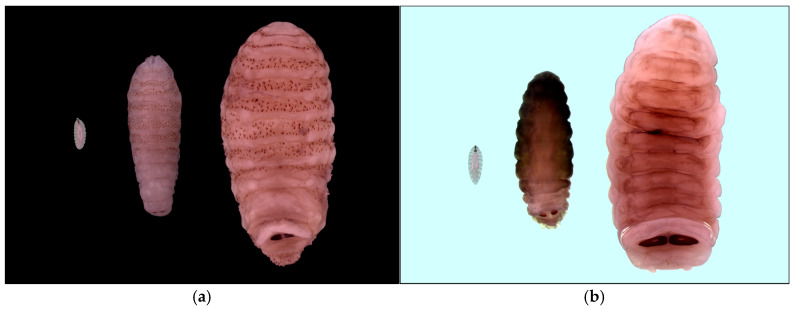
(**a**) Larval stages of *Cephenemyia stimulator* (L1 left, L2 center, and L3 right); (**b**) larval stages of *Oestrus ovis* (L1 left, L2 center, and L3 right).

**Figure 4 animals-15-02396-f004:**
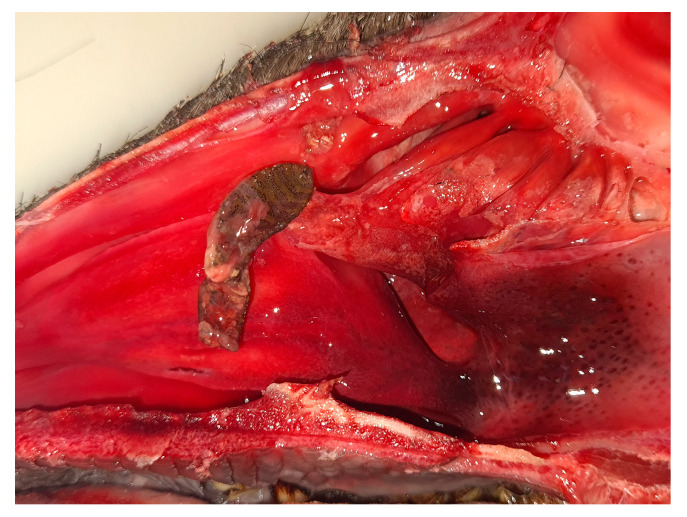
Partially decomposed *Cephenemyia stimulator* L3 removed from the maxillary sinus.

**Figure 5 animals-15-02396-f005:**
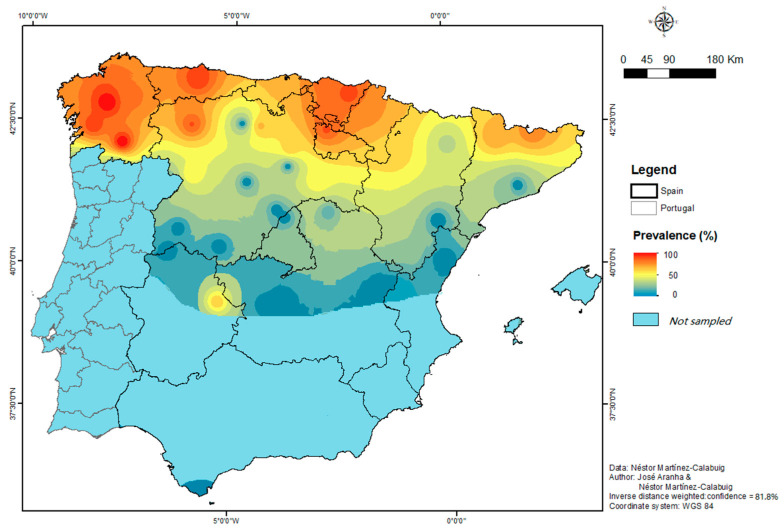
Spatial distribution of the prevalence of *Cephenemyia stimulator* in roe deer in the study area.

**Table 2 animals-15-02396-t002:** Factors that significantly influenced *Cephenemyia stimulator* prevalence.

Factor	Estimate	Z Value	Probability	OR	CI 95%
Month February ^1^	-	-	-	-	-
Month January	0.602	2.283	0.022	1.83	1.09–3.07
Month March	0.308	0.763	0.446	1.36	0.62–3.02
Month April	0.660	2.462	0.014	1.94	1.15–3.28
Month May	0.111	0.364	0.716	1.12	0.61–2.03
Month June	1.279	3.290	0.001	3.59	1.69–7.83
Month July	1.589	3.200	0.001	4.90	1.94–13.87
Month August	0.902	1.942	0.052	2.46	0.99–6.20
Month September	0.988	2.666	0.008	2.69	1.31–5.61
Month October	0.725	2.036	0.042	2.06	1.03–4.19
Month November	0.509	1.368	0.171	1.66	0.80–3.44
Month December	0.367	0.998	0.318	1.44	0.70–2.97
Year 2023 ^1^	-	-	-	-	-
Year 2018	−2.021	−2.513	0.012	0.13	0.03–0.70
Year 2019	−1.572	−2.629	0.009	0.21	0.06–0.68
Year 2020	−1.089	−1.949	0.051	0.34	0.11–1.02
Year 2021	0.188	0.550	0.582	1.21	0.62–2.38
Year 2022	0.763	3.201	0.001	2.14	1.35–3.43
Year 2024	0.449	2.221	0.026	1.57	1.06–2.33
Year 2025	0.836	2.875	0.004	2.31	1.31–4.09
AR Castilla-La Mancha ^1^	-	-	-	-	-
AR Andalucía	−14.230	−0.012	0.990	6.61 ×$\;10^{-7}$	−Inf – Inf
AR Aragón	1.400	1.911	0.056	4.05	1.02–18.69
AR Cantabria	3.512	3.420	<0.001	33.51	4.46–262.10
AR Castilla y León	3.156	4.999	<0.001	23.47	7.37–91.96
AR Cataluña	2.683	3.481	<0.001	14.62	3.41–72.41
AR Comunidad de Madrid	−13.766	−0.003	0.997	1.05 ×$\;10^{-6}$	- – Inf
AR Comunidad Foral de Navarra	3.885	4.418	<0.001	48.66	9.16–298.51
AR Comunidad Valenciana	−13.243	−0.015	0.988	1.77 ×$\;10^{-6}$	−Inf – Inf
AR Extremadura	20.274	0.017	0.987	6.38 × $\;10^{-8}$	−Inf – -
AR Galicia	5.084	4.690	<0.001	161.35	19.41–1416.05
AR La Rioja	5.122	6.783	<0.001	167.66	40.95–817.08
AR País Vasco	4.926	5.994	<0.001	137.80	29.79–772.16
AR Principado de Asturias	4.246	2.843	0.004	69.76	4.55–2174.45
Bio-region Interior Mountain + Age adult ^1^	-	-	-	-	-
Bio-region Atlantic + Age fawn	−19.196	−0.029	0.977	4.60 × $10^{-9}$	−Inf – Inf
Bio-region Atlantic + Age young	1.277	1.355	0.175	3.59	0.57–23.77
Bio-region Atlantic + Age adult	1.504	1.600	0.110	4.50	0.72–29.71
Bio-region Atlantic + Age old	2.172	2.055	0.040	8.77	1.14–74.37
Bio-region Northern Plateau + Age fawn	−16.916	−0.004	0.997	4.50	- – Inf
Bio-region Northern Plateau + Age Young	1.378	2.902	0.003	3.97	1.59–10.32
Bio-region Northern Plateau + Age adult	1.591	3.449	<0.001	4.91	2.02–12.45
Bio-region Northern Plateau + Age old	1.670	3.386	<0.001	5.31	2.05–1.43
Bio-region South Central + Age young	−15.881	−0.013	0.989	1.27 ×$\;10^{-7}$	- – Inf
Bio-region South Central + Age adult	−15.666	−0.020	0.984	1.57 ×$\;10^{-7}$	−Inf – Inf
Bio-region South Central + Age old	−14.007	−0.004	0.997	8.26 ×$\;10^{-7}$	- – Inf
Bio-region Interior Mountains + Age young	−0.471	−0.762	0.446	2.30	0.18–2.04
Bio-region Interior Mountains + Age old	0.787	1.378	0.168	2.20	0.71–6.84
Bio-region South and East Coast + Age young	−0.158	−0.148	0.883	3.14	0.08–6.15
Bio-region South and East Coast + Age adult	0.993	1.347	0.178	2.70	0.64–11.53
Bio-region South and East Coast + Age old	−13.908	−0.008	0.993	9.11 × $10^{-7}$	- – Inf
Land cover Agriculture ^1^	-	-	-	-	-
Land cover Broadleaf	0.848	3.101	0.002	2.34	1.37–4.01
Land cover Coniferous	1.236	4.261	<0.001	3.44	1.96–6.11
Land cover Mix forested areas	1.400	4.368	<0.001	4.06	2.18–7.65
Land cover Shrubs	1.307	5.291	<0.001	3.69	2.29–6.04
Abundance (0–0.3) ^1^	-	-	-	-	-
Abundance (0.3–0.7)	2.339	1.759	0.079	10.37	1.00–262.51
Abundance (0.7–1.4)	2.932	2.201	0.028	18.76	1.81–476.57
Abundance (1.4–2.6)	3.447	2.561	0.010	31.41	2.94–811.72
Abundance (2.6–9.8)	3.872	2.879	0.004	48.04	4.51–1239.53

^1^ Data that were used as a reference.

**Table 3 animals-15-02396-t003:** Factors that significantly influenced *Cephenemyia stimulator* mean intensity.

Factor	F Value	Df	*p* (>F)
Sex	11.446	1	<0.001
Age	42.504	2	<0.001
Year	4.706	7	<0.001
AR	7.674	10	<0.001

## Data Availability

According to the General Regulation on Data Protection (GRDP) regulated by Law 59/2019, all data treated within the scope of this paper are confidential.

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
