# Peer review of "Distribution of Nasal Myiases Affecting Roe Deer in Spain and Associated Risk Factors"

_animals, 2025, doi:10.3390/ani15162396_

Round 1

Reviewer 1 Report

Comments and Suggestions for Authors

The work entitled “Distribution of nasal myiasis affecting roe deer in the Iberian Peninsula and associated risk factors” submitted for evaluation is of great importance for understanding the epidemiology of myiasis and its relationship with climatic and environmental factors.

A significant number of samples have been analyzed; however, the sampling is not uniform throughout the study area, which could influence the results of the analyses. I believe the authors should explain why they decided to exclude the information from Portugal in the “roe deer abundance” variable of the predictive model.

If the reason is related to the small sample size (N=65), then data from Andalusia (N=10), the Community of Madrid (N=1), the Valencian Community (N=15), and Extremadura (N=2) should also be excluded, due to the low number of samples from these southern and eastern regions of the Peninsula.

From a biological point of view, organizing the information by administrative units makes little sense. It would be more appropriate to use biogeographical or geoclimatic criteria.

The authors should propose preventive and control measures to reduce roe deer density and, consequently, the prevalence of oestrosis.

It is recommended to develop a new predictive model with the recommendations provided.

In particular:

  • Line 170: Given the low number of samples from the southern and eastern parts of the Peninsula (Andalusia, Extremadura, Community of Madrid, and Valencian Community), they should have been excluded from the model.
  • Line 234: The 90% prevalence is found in Asturias, where the sample size is very small (N=10).
  • Lines 292–295: The authors argue that significantly higher prevalence was observed in forested and shrubland areas compared to agricultural areas composed of crops, pastures, and permanent plantations, which could be explained by a combination of factors making these areas favorable for parasite development. It is also very likely due to host selection of areas offering shelter, avoiding open areas due to fear or predation. It should be noted that in much of the northeastern Iberian Peninsula, wolves are present.

Author Response

The work entitled “Distribution of nasal myiasis affecting roe deer in the Iberian Peninsula and associated risk factors” submitted for evaluation is of great importance for understanding the epidemiology of myiasis and its relationship with climatic and environmental factors.

  • A significant number of samples have been analyzed; however, the sampling is not uniform throughout the study area, which could influence the results of the analyses. I believe the authors should explain why they decided to exclude the information from Portugal in the “roe deer abundance” variable of the predictive model.

Sampling in the different areas varies depending on their population density and the hunting permits granted by the different authorities responsible for controlling wildlife populations in each regional demarcation (different laws apply depending on the region). No information was available on the animals in Portugal regarding the variable "roe deer abundance" because the data used were obtained from the article by Fernández-López, J. et al. (2022), which only included data on roe deer abundance in Spain. However, Portugal was included in the logistic regression, and the statistical model itself decided to exclude the Portuguese animals from the final statistical model. Similarly, the data used on bioclimatic bioregions extracted from the "Spanish Wildlife Disease Surveillance Scheme (Internal report to the Spanish Ministry of Agriculture, MARM and spatial aggregation of wildlife. 2008)" also did not include Portugal. As a country with several distinct climatic zones from north to south, this made it difficult to assimilate it to the areas of Spain. Given this difficulty, it was decided not to include all animals in Portugal as belonging to a climatic zone, as this would not be true. Therefore, they were not assigned any climate zones, and the software did not include them in the model. This reason is unrelated to sample size, so we believe the remaining regions should be retained. In this case, well-defined climate zones and regions, even if only by political division, generate differences, as laws on wildlife control are controlled by regional government. Regarding biological/climatic variations, the climate zone factor was introduced into the initial model, and the statistical algorithm obtained the best mathematical result from the results set.

Since animals from Portugal could not be included in the model and may cause confusion, and according to with reviewer 3 indication, Portugal animales were removed from the manuscript, focusing only on animals from Spain, where all variables (including climatic bioregions and roe deer abundance) could be studied.

  • If the reason is related to the small sample size (N=65), then data from Andalusia (N=10), the Community of Madrid (N=1), the Valencian Community (N=15), and Extremadura (N=2) should also be excluded, due to the low number of samples from these southern and eastern regions of the Peninsula.

As we indicated in the previous point, the animals from Portugal were not eliminated from the model due to sample size, but because some of the variables that the algorithm considered most significant did not have a category assigned to the animals from Portugal. However, according to reviewer 3 suggestion , these animals were eliminated from the study.

  • From a biological point of view, organizing the information by administrative units makes little sense. It would be more appropriate to use biogeographical or geoclimatic criteria.

Thank you for this comment. Climatic bioregions of the "Spanish Wildlife Disease Surveillance Scheme (Internal report to the Spanish Ministry of Agriculture, MARM and spatial aggregation of wildlife. 2008)" were also use as one of the variables to be considered in the analysis. However, we believe that regional data (AR) can also be important, since each community is autonomous and has its own game regulations, and hunting conditions and seasons can vary greatly from one region to another.

  • The authors should propose preventive and control measures to reduce roe deer density and, consequently, the prevalence of oestrosis.

Thank you for your recommendation. A paragraph was added at the end of the discussion where we mention this point (lines 379-381)

It is recommended to develop a new predictive model with the recommendations provided. In particular:

  • Line 170: Given the low number of samples from the southern and eastern parts of the Peninsula (Andalusia, Extremadura, Community of Madrid, and Valencian Community), they should have been excluded from the model.

As explained in the first question, no variables were excluded based on the number of animals. Rather, the software itself (stepwise AIC consideration) excluded variables that could not interact in the model. This was the case in Portugal, as there was no data on roe deer abundance or climatic bioregion. For this reason, how it can lead to confusion, and under a reviewer indication, Portugal animals were removed from the study.

  • Line 234: The 90% prevalence is found in Asturias, where the sample size is very small (N=10).

I agree with the reviewer, although that sentence was only intended to summarize the range of values obtained across the entire territory which are subsequently broken down in the table 1. Although the sample size is low (high IC 95 because of a higher standard error), the algorithm discovers significance in relation to the region designated as a reference (Castilla La Mancha) which was chosen as a reference because it was the region that presented the lowest prevalence of those regions that had a representative sample size.

  • Lines 292–295: The authors argue that significantly higher prevalence was observed in forested and shrubland areas compared to agricultural areas composed of crops, pastures, and permanent plantations, which could be explained by a combination of factors making these areas favorable for parasite development. It is also very likely due to host selection of areas offering shelter, avoiding open areas due to fear or predation. It should be noted that in much of the northeastern Iberian Peninsula, wolves are present.

In accordance with the reviewer's requests, a sentence has been added to that paragraph of the discussion where reference is also made to the presence of wolves that may influence the habitat selection of roe deer. Lines 321-323.

Reviewer 2 Report

Comments and Suggestions for Authors

In this study, the authors analyzed the presence of the causative agent of nasal myiasis in roe deer, as well as the associated risk factors, across regions of Spain and Portugal. The results presented in this paper are of fundamental importance for implementing effective control and prevention measures for this disease in roe deer populations.

L120: Please reword this sentence. Suggested revision: “Figure 1 shows the geographic distribution of the collected samples.”

L133–135: Reword this sentence. For example, consider using the term “culled” instead of “hunted.”

L139: Instead of “based on”, use: “using Hoye’s dental criteria for age estimation.”

L172–173: Correct the sentence to: “Heads were transported either fresh or frozen (–20°C)…”

L211–216: Simplify this sentence. Suggested version: “Total prevalence in Spain (39.9%) was slightly higher than in Portugal (33.9%).”

L235–237: Please reword the sentence. For example: “Logistic regression identified seven significant predictors of C. stimulator prevalence.”

L238–240: The presentation of results here is unclear. In Table 1, the prevalence for the Low category (0.3–0.7) was slightly higher than for Medium (0.7–1.4), accompanied by a higher mean larvae intensity.

Figures: Please ensure consistent representation of longitude and latitude across all maps.

Author Response

In this study, the authors analyzed the presence of the causative agent of nasal myiasis in roe deer, as well as the associated risk factors, across regions of Spain and Portugal. The results presented in this paper are of fundamental importance for implementing effective control and prevention measures for this disease in roe deer populations.

  • L120: Please reword this sentence. Suggested revision: “Figure 1 shows the geographic distribution of the collected samples.”

Done. Lines 112-113

  • L133–135: Reword this sentence. For example, consider using the term “culled” instead of “hunted.”

Done. Lines 125-129

  • L139: Instead of “based on”, use: “using Hoye’s dental criteria for age estimation.”

It has been changed according to the reviewer's requests. Lines 144-145

  • L172–173: Correct the sentence to: “Heads were transported either fresh or frozen (–20°C)…”

It has been corrected. Line 132

  • L211–216: Simplify this sentence. Suggested version: “Total prevalence in Spain (39.9%) was slightly higher than in Portugal (33.9%).”

This sentence has been modified by removing the Portuguese animals from the study at the suggestion of a reviewer.

  • L235–237: Please reword the sentence. For example: “Logistic regression identified seven significant predictors of C. stimulator prevalence.”

    It has been rewritten according to the reviewer's requests. Line 228.

  • L238–240: The presentation of results here is unclear. In Table 1, the prevalence for the Low category (0.3–0.7) was slightly higher than for Medium (0.7–1.4), accompanied by a higher mean larvae intensity.

The reviewer is right. It has been written in a clearer way so as not to cause confusion. Lines 255-256

  • Figures: Please ensure consistent representation of longitude and latitude across all maps.

The figures (maps) have been modified so that they all have the same representation of longitude and latitude.

Reviewer 3 Report

Comments and Suggestions for Authors

Dear Authors,
I reviewed your manuscript "Distribution of nasal myiasis affecting roe deer in the Iberian Peninsula and associated risk factors."

The data you presented is interesting, but the manuscript needs major revision. See my comments below:

Lines 35 and 376: Similarly to line 30, use capital for "peninsula".

Abstract: The Animals' Instructions for Authors mentions: "The abstract should be a total of about 200 words maximum." Consequently, you should shorten your Abstract a bit, because it reaches 240 words.

Lines 48-51: I would remove this part of your statement: "...in areas with greater roe deer abundance and..." It makes perfect sense that the prevalence of a parasite would be greater in an area where the host of that parasite is abundant. I would keep: "The prevalence of C. stimulator was significantly higher in forested and scrubland habitats. In addition, young animals (> 3 months to 2 years old) showed notably higher larval intensity than adult and old roe deer, and no cases were detected in animals 3 months or younger."

Line 62: Use "the species" instead of " roe deer" to avoid its repetition from the previous sentence.

Line 65: Cut "the" before 26%.

Lines 67-70: You must rephrase this paragraph more clearly. You state that Cephenemyia infests cervids of the subfamilies Cervinae and Odocoileinae. At the same time, you state that four species are widespread in the Palearctic, including C. ulrichi. The latter primarily parasitizes the moose, included in the Capreolinae subfamily, while C. trompe infests the reindeer, of the same subfamily. The term "Odocoileinae" is considered a synonym of Capreolinae, the recognized taxon.

Line 72: Insert "they" before "quickly", otherwise it is understood that adults, not larvae, migrate into the nasal cavity.

Line 73: use the plural "L1s" that correlates with the plural "resume their lifecycle".

Line 76: "L3s"

Line 76: Cut "and" before "bury" becoming: "...coughs, bury..."

Line 84: Cut "between". It is unnecessary!

Lines 86-88: Reworded: "Furthermore, some mature L3 can occasionally become trapped in the nasal cavity, where they die and decompose, causing a purulent focus around them."

Line 89: "Oestrus is another Oestridae family" must become: "Oestrus is another genus of the Oestridae family".

Line 96 and 101: The correct form is "first", not "firstly"

Line 137: "These kinds of samples were..." or "This kind of sample was..."

Line 173: Use "where they were analyzed" instead of "where they analyzed".

Lines 211-226 (3.1. Overall prevalence and larval mean intensity subchapter): According to Carranza, J. 2010, "Ungulates and their management in Spain. In: Apollonio M, Andersen R, Putman R, editors. European Ungulates and their management in the 21st century. Cambridge (UK): Cambridge University Press; pp. 419–441", about 600000 roe deer live in Spain. By surveying 1600 specimens, you have examined approximately 0.26% of the entire population. Similarly, in Portugal, according to the Red Book of Mammals (2023), the estimated population is below 10,000 individuals. The 65 examined roe deer represent 0.65% of the total population. Comparing the percentages, you have examined approximately 3 times more roe deer in Portugal than in Spain. As such, the statement that the prevalence of these oestrids in Spain is higher than in Portugal may not be statistically valid. Please take my observation into account and modify your statements somewhat.

Line 239: "significantly", not "significant"

Lines 286-287: Use "in areas with higher host densities" instead of "in areas where host densities are higher". It sounds better!

Lines 295, 314-315, 324, and 352: Insert the reference number in brackets, after Morellet et al., Morrondo et al., Arias et al., Fidalgo et al., Martínez-Calabuig et al., and Fidalgo et al. (2023), similarly to Kiraly and Egri [41] (lines 302/303). Don’t use italics for “et al.”

Lines 369-373: The two parts of this lengthy statement are redundant. Please modify the sentence.

Conclusions section (lines 369-378): Your conclusions are actually repetitions of the results obtained, but without figures. This section needs to be rewritten to state more synthetic conclusions.

General comment: I understand your desire for this study to cover the entire Iberian Peninsula. However, the small number of animals examined in Portugal (although percentage-wise higher than in Spain) and the exclusion of roe deer collected in Portugal from the calculation of two variables lead me to suggest that the data relating to Portugal be removed from the study/manuscript and that only those obtained in Spain be retained. Hope you agree.

Question: Did the advanced degree of decomposition of the dead larvae collected from the maxillary sinuses allow their identification based solely on morphological criteria? Do you consider that their molecular identification was not necessary?

Comments on the Quality of English Language

Several grammar and typo elements (missing articles, commas, sentences that are too long, etc.) require revision.

Author Response

The data you presented is interesting, but the manuscript needs major revision. See my comments below:

  • Lines 35 and 376: Similarly to line 30, use capital for "peninsula".

We have used capitals for Peninsula along the text; however, as Portugal animalas have been eliminated from the study, some Iberian Peninsula were changed to peninsular Spain, and adjectival structure.

  • Abstract: The Animals' Instructions for Authors mentions: "The abstract should be a total of about 200 words maximum." Consequently, you should shorten your Abstract a bit, because it reaches 240 words.

Thanks for the suggestion. The abstract has been adjusted to the maximum of 200 words.

  • Lines 48-51: I would remove this part of your statement: "...in areas with greater roe deer abundance and..." It makes perfect sense that the prevalence of a parasite would be greater in an area where the host of that parasite is abundant. I would keep: "The prevalence of C. stimulator was significantly higher in forested and scrubland habitats. In addition, young animals (> 3 months to 2 years old) showed notably higher larval intensity than adult and old roe deer, and no cases were detected in animals 3 months or younger."

The abstract has been restructured to be clearer and to fit the maximum of 200 words; many phrases have changed completely.

  • Line 62: Use "the species" instead of " roe deer" to avoid its repetition from the previous sentence.

Thank you for the comment. Changed in line 59

  • Line 65: Cut "the" before 26%.

This sentence was removed from the revised version due to the exclusion of animals from Portugal

  • Lines 67-70: You must rephrase this paragraph more clearly. You state that Cephenemyia infests cervids of the subfamilies Cervinae and Odocoileinae. At the same time, you state that four species are widespread in the Palearctic, including C. ulrichi. The latter primarily parasitizes the moose, included in the Capreolinae subfamily, while C. trompe infests the reindeer, of the same subfamily. The term "Odocoileinae" is considered a synonym of Capreolinae, the recognized taxon.

Thank you. The correct term should have been Capreolinae instead of Odocoileinae. In addition, the entire sentence has been reworded to avoid confusion. Lines 62-65

  • Line 72: Insert "they" before "quickly", otherwise it is understood that adults, not larvae, migrate into the nasal cavity.

Done. A period was added and L1 was specified to avoid confusion. Line 67

  • Line 73: use the plural "L1s" that correlates with the plural "resume their lifecycle".

Added. Line 69

  • Line 76: "L3s"

Added. Line 71

  • Line 76: Cut "and" before "bury" becoming: "...coughs, bury..."

Done. Line 72

  • Line 84: Cut "between". It is unnecessary!

Done

  • Lines 86-88: Reworded: "Furthermore, some mature L3 can occasionally become trapped in the nasal cavity, where they die and decompose, causing a purulent focus around them."

It has been reworded according to the reviewer's request. Lines 81-83

  • Line 89: "Oestrus is another Oestridae family" must become: "Oestrus is another genus of the Oestridae family".

Changed. Thank you. Line 84

  • Line 96 and 101: The correct form is "first", not "firstly"

Corrected. Line 91

  • Line 137: "These kinds of samples were..." or "This kind of sample was..."

During the restructuring of the text this sentence was removed

  • Line 173: Use "where they were analyzed" instead of "where they analyzed".

This text has been changed like the previous one in the reviewed version

  • Lines 211-226 (3.1. Overall prevalence and larval mean intensity subchapter): According to Carranza, J. 2010, "Ungulates and their management in Spain. In: Apollonio M, Andersen R, Putman R, editors. European Ungulates and their management in the 21st century. Cambridge (UK): Cambridge University Press; pp. 419–441", about 600000 roe deer live in Spain. By surveying 1600 specimens, you have examined approximately 0.26% of the entire population. Similarly, in Portugal, according to the Red Book of Mammals (2023), the estimated population is below 10,000 individuals. The 65 examined roe deer represent 0.65% of the total population. Comparing the percentages, you have examined approximately 3 times more roe deer in Portugal than in Spain. As such, the statement that the prevalence of these oestrids in Spain is higher than in Portugal may not be statistically valid. Please take my observation into account and modify your statements somewhat.

In accordance with the reviewer's final request, it has been decided to remove the animals from Portugal from the manuscript, so this point would be resolved.

  • Line 239: "significantly", not "significant"

Modified. Line 253

  • Lines 286-287: Use "in areas with higher host densities" instead of "in areas where host densities are higher". It sounds better!

This sentence has been removed when rewriting this part in the discussion

  • Lines 295, 314-315, 324, and 352: Insert the reference number in brackets, after Morellet et al., Morrondo et al., Arias et al., Fidalgo et al., Martínez-Calabuig et al., and Fidalgo et al. (2023), similarly to Kiraly and Egri [41] (lines 302/303). Don’t use italics for “et al.”

Done. Line 300, 311, 315, 323

  • Lines 369-373: The two parts of this lengthy statement are redundant. Please modify the sentence.

Conclusions have been completely rewritten.

  • Conclusions section (lines 369-378): Your conclusions are actually repetitions of the results obtained, but without figures. This section needs to be rewritten to state more synthetic conclusions.

The conclusions section has been rewritten to provide more concise information, avoiding repetition of results.

  • General comment: I understand your desire for this study to cover the entire Iberian Peninsula. However, the small number of animals examined in Portugal (although percentage-wise higher than in Spain) and the exclusion of roe deer collected in Portugal from the calculation of two variables lead me to suggest that the data relating to Portugal be removed from the study/manuscript and that only those obtained in Spain be retained. Hope you agree.

We decided to comply with the reviewer's request because the animals from Portugal were removed by the statistical model software due to the lack of data for two of the variables studied (bioregions and roe deer abundance). This could have led to confusion in the final results, so it was ultimately decided to exclude these animals from the study. Thank you for your recomendation.

  • Question: Did the advanced degree of decomposition of the dead larvae collected from the maxillary sinuses allow their identification based solely on morphological criteria? Do you consider that their molecular identification was not necessary?

Yes, because although the entire contents of the larva decompose, the cuticle, which is largely made of chitin, resists without disintegrating, allowing us to observe the pattern of spines, antennal lobes and/or respiratory plates that allow us to distinguish them morphologically. A short sentence has been added to clarify this point for readers. Lines 213-214

  • Several grammar and typo elements (missing articles, commas, sentences that are too long, etc.) require revision.

Thank you for your recommendation. The entire manuscript has been reviewed by a professional translator.

Round 2

Reviewer 1 Report

Comments and Suggestions for Authors

The paper titled “Distribution of nasal myiasis affecting roe deer in the Iberian Peninsula and associated risk factors” submitted for review is of great importance for understanding the epidemiology of myiasis and its relationship with climatic and environmental factors.

The authors have provided a series of explanations and have revised the manuscript in a new version, including the suggested changes, which in my opinion has improved its accuracy and quality, making it suitable for publication as is.

Reviewer 3 Report

Comments and Suggestions for Authors

Dear Authors,
I revised your reviewed manuscript "Distribution of nasal myiasis affecting roe deer in the Iberian Peninsula and associated risk factors."

I noticed that you restricted the data presented in the manuscript to Spain. I am glad you accepted this request because the data from Portugal lacked statistical value, giving the study a feeling of superficiality.

You also rewrote the conclusions, removing that aspect of the results' repetition. At the same time, you accepted my other suggestions, thus improving the quality of the manuscript. I believe it can be published in its current form.